# Perspectives and experiences with sleep and recovery among women receiving buprenorphine for opioid use disorder

**Michelle Eglovitch**[1]*, **Anna Beth Parlier-Ahmad**[1], **Alison J. Patev**[1], **Brenna Cook**[2], **Chengxian Shi**[2], **Stephanie Violante**[1], **Joseph M. Dzierzewski**[3], **Morgan H. James**[4,5,6,7], **Caitlin E. Martin**[8,9]

**1** Department of Psychology, Virginia Commonwealth University, Richmond, Virginia, United States of America, **2** School of Medicine, Virginia Commonwealth University, Richmond, Virginia, United States of America, **3** National Sleep Foundation, District of Columbia, Washington, United States of America, **4** Department of Psychiatry, Robert Wood Johnson Medical School, Rutgers University, Piscataway, New Jersey, United States of America, **5** Rutgers Addiction Research Center, Brain Health Institute, Rutgers Health, Piscataway, New Jersey, United States of America, **6** School of Psychology, Faculty of Science, University of Sydney, Sydney, New South Wales, Australia, **7** Brain and Mind Centre, The University of Sydney, Sydney, New South Wales, Australia, **8** Institute for Drug and Alcohol Studies, Virginia Commonwealth University, Richmond, Virginia, United States of America, **9** Department of Obstetrics and Gynecology, School of Medicine, Virginia Commonwealth University, Richmond, Virginia, United States of America

* eglovitchms@vcu.edu

## Abstract

Sleep issues are prevalent among women receiving medication for opioid use disorder (OUD). However, there is limited data about subjective sleep experiences and how they relate to OUD trajectories. This mixed-methods study explored the intersection of sleep and OUD recovery from the patient perspective among a sample of women receiving medication for OUD. This study enrolled non-pregnant women aged 18-65 who were stabilized on buprenorphine from an outpatient OUD program. Participants were recruited during their routine treatment visits, and enrollment occurred from February 2022 through September 2023. Study participants who endorsed clinically elevated insomnia symptoms on the Insomnia Severity Index (ISI) (≥11 score) (n=50) were included in current study analyses. A sub-sample (n=11) who met the ISI threshold participated in semi-structured interviews. Survey responses were analyzed using descriptive statistics, and interviews were analyzed using applied thematic analysis. The average length of time on buprenorphine for the overall sample was 30 months (range: 2 months – 245 months). Participants reported engagement in healthy sleep behaviors, grouped into four domains: positive sleep related cognitions, sleep environment, sleep restriction, and reducing stimulating activities. Respondents characterized the multidimensional relationship between sleep and health. Women also described how sleep evolves through addiction into recovery, and how good sleep reduces risk of return to substance use. Finally, women discussed the impacts that medication for OUD have on sleep, specifically how they might time their buprenorphine to align with sleep and how it might impact their energy levels. We found that sleep is a dynamic process among this sample of women receiving medication for OUD. Findings are intended to inform future investigations of the mechanisms underlying

**Data availability statement:** All quantitative data files are available via Open Science Framework Project. The de-identified raw datasets are currently available to review at: https://osf.io/kvj2x/.

**Funding:** This study is supported by NIDA (K23 DA053507 to CEM, R00 045765 to MHJ, R01 061303 to MHJ, T32DA007027 to ME), and the National Center for Advancing Translational Sciences (UM1TR004360 to VCU). The funders had no role in study design, data collection and analysis, decision to publish, or preparation of the manuscript.

**Competing interests:** The authors have declared that no competing interests exist.

the sleep-OUD intersection. In addition, this study reflects the importance of incorporating patient perspectives into the development of therapeutics targeting this patient population.

## 1. Introduction

The opioid crisis is a serious public health issue in the United States and has brought an increasing awareness to the societal impacts of opioid use disorder (OUD) [1]. Medication for opioid use disorder (MOUD) is the gold standard treatment for OUD to improve mortality, morbidity, and overall health outcomes for this population. MOUD, including buprenorphine and methadone, works as an opioid agonist and helps facilitate recovery in people with OUD. MOUD accomplishes this by alleviating withdrawal symptoms and cravings, leading to decreased illicit substance use and improved social functioning and quality of life [2,3]. Many people receiving MOUD continue the medication long-term as treatment for their OUD. Although MOUD is a life-saving treatment, drop out, discontinuation rates, and subsequent risk of return to substance misuse remain high in this population [4]. One multi-state study of 17,329 Medicaid patients found that over 25% of the sample discontinued in the first month of treatment, and more than half the sample discontinued before six months [5]. Co-morbid health issues such as psychological and psychiatric conditions may contribute to such high drop-out rates; thus, elucidating common co-morbidities may help improve the quality of treatments for OUD [6].

Poor sleep, or difficulty falling and/or staying asleep, is a prominent co-occurring health issue among individuals receiving MOUD. Sleep issues are highly prevalent among people receiving MOUD; one systematic review found prevalence rates of sleep disorders in this population ranged from 19-41%, with one recent study finding clinically significant insomnia symptoms in up to 60% of the sample [7,8]. In addition, the consequences of poor sleep in this population are critical. For those with substance use disorders (SUDs) who are in recovery, research demonstrates that insomnia increases risk of return to use, even years into treatment [9,10].

Research demonstrates that among individuals receiving MOUD, having sleep issues may increase risk for substance use recurrence via multiple mechanisms [11,12]. One pathway is that poor sleep leads to increased impulsivity and stress reactivity, both of which may subsequently increase risk of return to non-prescribed substance use [13,14]. In addition, the relationship between sleep and OUD might further be complicated by being on MOUD, as preclinical models demonstrate that buprenorphine treatment is associated with increased time spent awake and increased sleep latency [15].

Further, the multi-faceted relationship between OUD and sleep likely varies between men and women. Certain sleep conditions such as insomnia are more prevalent among women than men [16]. In the context of OUD, women present to treatment with more comorbid conditions than men, and women experience more severe impairment related to social, occupational, medical and psychiatric functioning associated with OUD compared to men [17,18]. There are multiple proposed reasons for these gendered intricacies, including the impact of social determinants of health on sleep that more commonly affect women, such as caregiving, as well as sex-based variables, such as hormonal fluctuations [19–21]. Thus, addressing sleep may be an avenue to improve quality of OUD treatment through a sex and gender informed lens.

Although sleep issues and OUD are often comorbid, there is a lack of published data from the patient perspective on the experiences of individuals with OUD who have sleep issues. This reflects a larger issue in substance use research, which is built upon a quantitative

evidence base despite the fact that substance use behavior is largely socially constructed [22]. Thus, qualitative data can help researchers understand the contextual and social factors that relate to substance use disorders, particularly with specific populations such as women with OUD. Qualitative insights can reveal treatment gaps for this population by delving into experiences, perspectives, and motivations for engaging with the healthcare system. In addition, qualitative data are not constrained by a priori hypotheses, as with quantitative studies, and thus can help inform clinical investigations to ensure that lived experiences are considered in the development of patient-centered interventions.

To our knowledge, qualitative studies have not yet evaluated women's perspectives of how sleep and OUD recovery intersect while receiving medication treatment for their OUD [23]. In the context of sleep, subjective sleep experiences uniquely contribute to the evidence base informing sleep health therapeutics, as they provide insights into the extent to which patients may be experiencing sleep that may not otherwise be detected by providers [24]. Thus, collecting subjective experiences with sleep can be a critical first step in understanding how people on MOUD may experience sleep differently than other populations. These patient-reported outcomes can help researchers and clinicians understand how sleep interventions can be developed or tailored and subsequently improve treatment in the OUD clinical setting. This study was conducted among a sample of women receiving buprenorphine for OUD to begin to answer the question, "What have been your experiences with sleep through the evolution of addiction into long-term recovery on MOUD?"

## 2. Materials and methods

All study procedures were approved by the Virginia Commonwealth University Institutional Review Board (HM20023390). Written consent was obtained from all participants.

### 2.1 Study design and participants

This mixed-methods study sought to explore sleep from the perspective of women on MOUD in an outpatient addiction treatment clinic. Clinic patients are in differing stages of recovery and use a wide range of wraparound services from the clinic, including addiction medicine specialists, social work, and behavioral therapy. Participants of this study were female and not pregnant nor within 6 weeks postpartum, aged 18-65, English-speaking, and stabilized on buprenorphine for OUD for at least 6 weeks. Patients were excluded if they demonstrated serious comorbid cognitive or psychiatric impairment, as determined based on chart review, trained clinical judgment of the research coordinator, or an inability to complete informed consent procedures. Enrollment occurred from February 2022 through September 2023 (recruitment rate: 76.0%).

The research coordinator pre-screened potentially eligible participants by medical chart review then approached patients in the clinic upon arrival for their routine OUD visits to invite them to participate in the survey component of the study. Participants were also informed that they may be asked to participate in an additional component of the study, which would comprise of an hour-long interview. Participants completed a REDcap electronic survey in-person in the clinic on a tablet or a clinic computer, or remotely on personal devices based on participant preference. The research coordinator was not part of the treatment center team, and participants were assured throughout the consenting process that lack of participation would not impact their clinical care. Participants were reimbursed $30 for study completion.

Study staff purposively sampled the participants by identifying survey participants who endorsed clinically elevated insomnia symptoms as they completed the survey portion of the

study. After they completed the survey portion, the research coordinator invited them by call or text to participate in a semi-structured interview (up to three invitation attempts). Participants were reimbursed an additional $30 for interview completion. Recruitment and conduct of interviews continued until data saturation occurred, at which point recruitment for the interview portion of the study was completed (n=11). Specifically, the research coordinator and PI determined that thematic saturation was reached and no new information would be obtained from additional interviews [25–27]. Finally, medical record data indicating treatment history was abstracted and entered into REDcap for all participants who consented to being part of the study.

Seventy-eight participants enrolled in the study. The current analyses encompass survey data from participants who reported clinically elevated insomnia symptoms (n=50) and qualitative data from participants who completed the semi-structured interviews (n=11). Analyses regarding differences between participants who did and did not report insomnia symptoms in the larger study are found elsewhere [8].

## 2.2  Measures

**2.2.1 Quantitative.**  Demographic survey items included age and race. Psychosocial survey items included employment, education, and marital status. Treatment items including length of time in treatment, buprenorphine dose, and insurance status were collected via chart abstraction.

The Insomnia Severity Index (ISI) evaluated insomnia symptoms [28]. The ISI includes seven items utilizing a scale from 0- none to 4- very severe, for an overall score ranging from 0-28. Higher scores on the ISI indicate higher levels of insomnia symptoms. We used cut-offs of ≥11 to identify clinically elevated insomnia symptoms [28].

In addition, women were asked about their sleep health behaviors and how often they engaged in them on a 4-point Likert scale [0-rarely or not at all to 3-consistently]. The team intended to explore the prevalence of such behaviors as rooted in the clinical experiences; as such, these items were created by the research team, who have clinical and research expertise in the intersection between sex and gender, sleep, and OUD. Further, the questions were informed by conceptual frameworks that highlight gender variables in research [29]. For presentation purposes, we grouped these sleep health survey items into four categories and report their mean scores: (a) positive sleep related cognitions, including acceptance of a night of insufficient sleep (2 items); (b) healthy sleep environment, including use of a dark and cool room (3 items); (c) sleep restriction behavior, including limiting bed for sleep and sex (4 items); and (d) reducing stimulating activities at night including screen time, caffeine, and exercise (3 items). Participants also reported on how many times they got up throughout the night, and the reasons for getting up.

The General Anxiety Disorder-7 (GAD-7) evaluated anxiety symptoms. The GAD-7 is a validated tool to measure anxiety symptomology over the past 2 weeks, measuring 7 items on a scale from 0-not at all to 3-nearly every day, for an overall score ranging from 0-21. Higher scores on the GAD-7 indicate higher levels of anxiety symptomology. For the purposes of this study, scores ≥10 were considered reflective of clinically significant anxiety symptoms [30].

The Patient Health Questionnaire-9 (PHQ-9) evaluated depression symptoms. The PHQ-9 is a validated tool to measure depressive symptomology over the past 2 weeks, measuring 9 items on a scale from 0-not at all to 3-nearly every day, for an overall score ranging from 0-27. Higher scores on the PHQ-9 indicate higher levels of depressive symptoms. For the purposes of this study, scores ≥10 were considered reflective of clinically significant symptoms of depression [31].

The Patient-Reported Outcomes Measurement Information System- Global Health (PROMIS-10) assessed self-reported physical, mental and social health, including symptoms such as pain, function and general perceptions of health and wellbeing [32]. The PROMIS-10 is a 10-item measure that uses a 5-point Likert scale, with scale items varying depending on the individual question. We report scores for individual PROMIS-10 items.

Finally, the Distress Tolerance Scale (DTS) measured perceived capacity to tolerate distress. The DTS measures 4 domains: ability to tolerate emotions, assessment of the emotional situation as acceptable, level of attention absorbed by the negative emotion, and ability to regulate emotion [33]. The DTS is a 16-item measure using a scale from 1- strongly agree to 5- strongly disagree. Higher scores on the DTS indicate greater levels of distress tolerance.

**2.2.2 Qualitative.** A multidisciplinary team convened (addiction physician, psychologist, sleep expert) and developed the in-depth interview guide. The guide utilized open ended questions with prescribed prompts to explore patients' perspectives on sleep and how sleep relates to OUD recovery. Women were asked about qualities of healthy sleep, circumstances in their lives that impacted their ability to sleep, and the evolution of their sleep over recovery as well as the role of MOUD in their sleep.

The research team chose to collect qualitative data using individual semi-structured interviews instead of focus groups, given participant preferences for this structure in prior studies conducted at the same study site [34]. Study staff provided participants with the option of completing interviews using Zoom, by phone, or in person in a private office space at the outpatient addiction clinic. The research coordinators (ME + SV) conducted the interviews, and at least one research assistant informally took notes during interviews to follow up for clarity or any missed questions as needed. Zoom recorded and automatically transcribed the interviews. After transcript generation, the research assistant listened to each interview in conjunction with reading the transcript to check for fidelity, editing the transcript as needed for clarity. For in-person and phone interviews (n=3), an encrypted audio recording phone application was used, and the research assistant transcribed the interview. Both Zoom and in-person/phone transcript checks resulted in a final version in Microsoft Word.

**2.2.3 Researcher's positionality.** With qualitative research, it is critical to understand the positionality of the coding team and, therefore, the lens in which the team views the data. The coding team does not have the shared lived experience of OUD of the participants. Thus, the authors were cognizant of their own positionality and ensured the study was sensitive to the setting in which it was conducted.

## 2.3 Data management and analysis

The study team analyzed interview data using a multi-step process. Once interviews were transcribed, the qualitative team (ME, BC, CS, and CEM) each reviewed each of the 11 transcripts twice and assigned preliminary codes to the data. The study team then met regularly to combine and refine the list of codes, then grouped the finalized codes into themes. Coding and analysis was done using NVivo 14 software (QSR International, 2023). More detailed information about these study methods are described elsewhere [35].

This analysis used an inductive content analysis approach to the qualitative data, which meant that data was analyzed without preconceived notions on themes [36,37]. Thus, members of the research team reviewed each interview in its entirety, allowing the codes, grouping of codes, and subsequent themes to naturally arise from the participants words. Inductive content analysis was chosen as our approach as we wanted to view the data without assumption on what it might hold, and we sought to identify patterns from the data as they emerged [27]. This approach facilitated a deeper understanding of the sleep experiences for women in treatment for OUD.

This study utilized a sequential mixed methods design, meaning that based on the qualitative findings, we selected quantitative data for analysis from our robust battery of measures [38]. For example, as sleep health behaviors naturally arose from the qualitative data, we chose to analyze quantitative data around such behaviors as collected in the survey battery. Descriptive statistics and univariate analyses were produced for demographic, psychosocial, clinical, and these self-reported sleep health behavior variables using SPSS version 28. Such statistics were generated for participants with clinically elevated insomnia symptoms (n=50) and qualitative participants (n=11). Then, qualitative data was chosen to be presented by naming subthemes and overarching themes and assigning quotes to represent the individual sub-themes. Thus, both quantitative and qualitative data are presented in the results below to provide a comprehensive picture of the study sample.

## 3. Results

### 3.1 Quantitative

Of the 78 enrolled study participants, 50 participants (64.0%) reported clinically elevated insomnia symptoms. Of these 50 participants, 11 participated in a qualitative interview (Table 1). Qualitative participants' sociodemographic characteristics were similar to those of the insomnia sample from the larger survey study. Among the 11 participants in the qualitative sub-sample, the majority were white, unemployed, and unmarried.

Table 2 demonstrates the clinical characteristics of participants. The average length of time on buprenorphine for the overall sample was 30 months (range: 2 months – 245 months). Anxiety and depression were highly prevalent in the overall sample, with nearly half of participants endorsing clinically significant anxiety, and almost 3/4ths of the sample endorsing

**Table 1.  Sociodemographic characteristics among respondents with clinically elevated insomnia symptoms.**

| Sociodemographic Characteristic | Clinically elevated insomnia symptoms N (%) n = 50 | Qualitative participants N (%) n = 11 |
|---|---|---|
| Age [M (SD)] | 37.8 (9.2) | 35.2 (5.9) |
| Race | | |
| White | 32 (64.0%) | 9 (81.8%) |
| Black | 15 (30.0%) | 2 (18.2%) |
| Other | 3 (6.0%) | 0 (0%) |
| Employment | | |
| Unemployed | 25 (50.0%) | 8 (72.7%) |
| Employed | 15 (30.0%) | 3 (27.3%) |
| Disabled | 10 (20.0%) | 0 (0%) |
| Education | | |
| <High School | 21 (42.0%) | 5 (45.5%) |
| > High school | 29 (58.0%) | 6 (54.5%) |
| Marital status | | |
| Unmarried | 29 (58.0%) | 7 (63.6%) |
| Married | 21 (42.0%) | 4 (36.4%) |
| Insurance status | | |
| Medicaid/Medicare | 45 (90.0%) | 10 (90.9%) |
| Private insurance | 3 (6.0%) | 1 (9.1%) |
| Uninsured | 2 (4.0%) | 0 (0%) |

Table 2. Clinical Characteristics.

| Clinical Characteristic | Clinically elevated insomnia symptoms N (%) n = 50 | Qualitative participants N (%) n = 11 |
|---|---|---|
| Mental health | | |
| Clinically significant anxiety symptoms (% yes) | 23 (46.0%) | 4 (36.4%) |
| Clinically significant depressive symptoms (% yes) | 35 (70.0%) | 6 (54.5%) |
| Co-morbid clinically significant anxiety and depressive symptoms (% yes) | 23 (46.0%) | 4 (36.4%) |
| Average GAD-7 score (mean [SD]) (range: 0-21) | 10.5 (6.5) | 8.9 (4.9) |
| Average PHQ-9 score (mean [SD]) (range: 0-27) | 13.7 (6.5) | 12.1 (6.7) |
| Physical health | | |
| Self-reported physical health (% good/excellent) | 26 (52.0%) | 5 (45.4%) |
| Average pain (mean [SD]) (range: 0-10) | 6.0 (2.6) | 6.0 (1.8) |
| Treatment characteristics | | |
| Length of time on buprenorphine (months) (mean [range]) | 29 (2-245) | 43 (2-245) |
| Buprenorphine dose (milligrams) (median [range]) | 24 (4-30) | 24 (8-24) |
| History of prior opioid overdose (% yes) | 22 (44.0%) | 4 (36.4%) |
| Distress tolerance | | |
| Overall (mean [SD]) (range: 0-5) | 3.3 (.8) | 3.3 (.6) |
| Domain 1: Tolerance (range: 0-5) | 3.4 (1.0) | 3.3 (1.1) |
| Domain 2: Absorption (range: 0-5) | 3.4 (1.0) | 3.2 (1.1) |
| Domain 3: Appraisal (range: 0-5) | 3.1 (0.9) | 3.2 (.8) |
| Domain 4: Regulation (range: 0-5) | 3.2 (0.9) | 3.4 (.6) |

clinically significant depression. In addition, nearly half of participants in both categories endorsed co-morbid anxiety and depressive symptoms. The median buprenorphine dose was 24mg/day, which is within the standard range of buprenorphine dosing.

Participants also reported on healthy sleep behaviors, which were grouped into four domains: positive sleep related cognitions, healthy sleep environment, sleep restriction behavior, and reducing stimulating activities at night (Fig 1). The most highly endorsed healthy sleep behavior domain was a healthy sleep environment, and the least commonly endorsed healthy sleep behavior domain was reducing stimulating activities before bed.

Of the 50 participants, 35 (70.0%) endorsed getting up 3 or more times during the night, and the remainder of participants (n=15; 30.0%) endorsed getting up 1 or 2 times throughout the night. When asked the reasons for getting up during the night, over half the participants endorsed anxiety or stress (n=28; 56.0%) and physical discomfort (n=26; 52.0%).

## 3.2 Qualitative

Overall, all women that participated in the qualitative interviews indicated that sleep was an important component of health and OUD recovery. Participants expressed that sleep affects their mood; women mentioned that lack of sleep made them feel irritable and/or cranky, impatient, and overall affects their mood negatively. Several women (n=3) indicated that lack of sleep caused their anxiety to worsen. Women noted specific qualities of healthy sleep, ranging from feeling rested and refreshed (n=7), getting at least 6 hours of sleep (n=7), feeling motivated and productive (n=6), and uninterrupted sleep (n=6). Three themes emerged: 1)

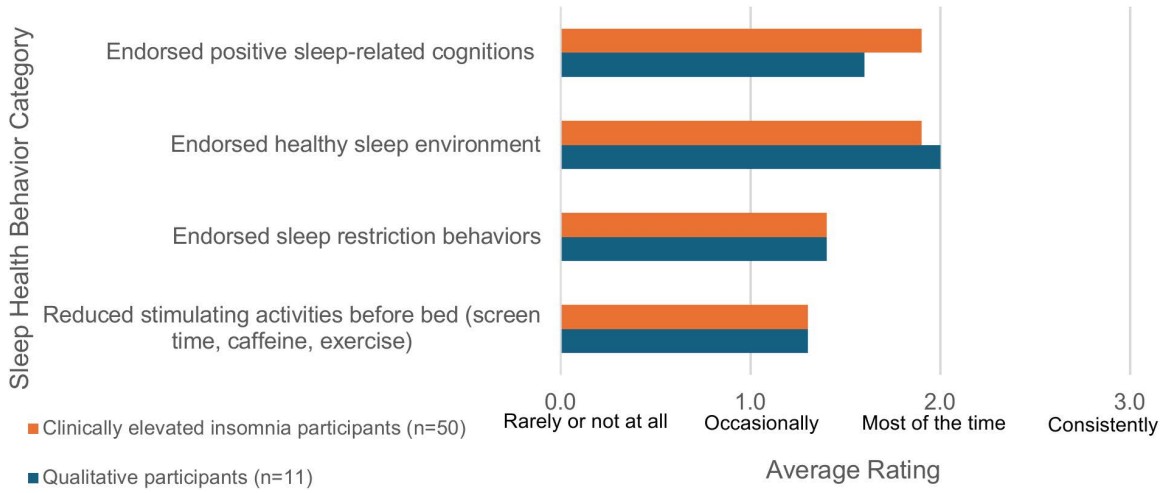

**Fig 1. Healthy Sleep Behaviors.**

Multidimensional factors impact sleep; 2) Sleep and journey of recovery, and 3) Buprenorphine's role in sleep. Table 3 lists the themes and subthemes, representative quotes.

**Theme 1: Multidimensional factors impact sleep.** Throughout the interviews, women shared about their complex lives and environments that affect their sleep. They discussed factors that have had a positive impact on their sleep including behavioral strategies, prescribed and non-prescribed substances, and environmental influences. All qualitative participants reported that engaging in positive sleep behaviors helps to improve their sleep (n=11), including reducing exercise before bed, and having an established bedtime routine. Some women (n=5) also shared that they employ positive coping skills, such as mindfulness, meditation, and breathing exercises. In addition, women mentioned that some medications have been helpful for sleep, such as melatonin (n=4), trazadone (n=3), hydroxyzine (n=2), and lorazepam (n=1). Finally, women highlighted helpful community factors, such as having family at home and being in a safe environment, that contribute to better sleep outcomes.

Women also described a multitude of factors that negatively impact their sleep, such as pregnancy, postpartum, and caregiving, physical and mental health, and environmental conditions. For example, most women (n=10) identified as mothers and shared that being pregnant and postpartum was detrimental to their sleep, sharing that the discomfort of pregnancy in the third trimester negatively affected their ability to sleep. Relatedly, women shared how caregiving for children or other family members also impacted their sleep. They described waking up to give care, as well as how their sleep schedules centered around giving care. In addition, women described a cyclical nature of mental and physical health in sleep, in that these factors would impact their ability to get good sleep, and then the lack of sleep would subsequently affect their physical and mental health. Finally, just as women shared how having a positive environment helps facilitate sleep, they also described how sharing a sleeping space with others, being in a noisy or hot environment, or not feeling safe is associated with poorer sleep outcomes.

**Theme 2: Sleep and journey of recovery.** Most participants described the evolution of sleep in three different stages: over the process of active substance use, through early treatment, and into long-term recovery. Women described how disrupted their sleep was in active addiction (n=10), with participants describing how they felt like they were sleeping all the time, with some reporting that they felt they weren't sleeping at all. They then shared

**Table 3.** Qualitative themes, sub-themes and quotes.

| Theme | Sub-theme | Quotation |
|---|---|---|
| Multidi-mensional factors impact sleep | Positive impact (behavioral strategies, prescribed and non-prescribed substances, environmental influences) | "Grounding [has been helpful] with sleep. Like on the times when I do feel like I'm withdrawing or something in my muscles, I'll do the one where you'll like squeeze your face as hard as you can and then let it go. Squeeze every piece of your body and it feels good after that." -Study ID 057 |
| | Negative impact (Caregiving, pregnancy, postpartum, mental and physical health, environmental conditions) | "It's already difficult for addicts to sleep, then having a baby on top of that. I went into treatment shortly after I found out that I was pregnant. I was uncomfortable towards the end of the pregnancy and then once the baby came then it was, you know, it's already hard for a mom that's not struggling with addiction. So, when you're in recovery, if you're not going to meetings and not meeting with your sponsor then you're not going to be able to stay clean because you're not sleeping. Even without a baby, it's going to be hard to sleep. And then adding a baby in the mix is very difficult." -Study ID 21 |
| Sleep and journey of recovery | Sleep evolves through addic-tion into long-term recovery | "I'm actually sleeping somewhat now. Me on dope and in the streets, is like me right now times a million. There is no such thing as sleep. And that's how my sleep has changed. At least I'm sleeping somewhat now. When… I was on drugs… there was nothing else but that." -Study ID 34 |
| | Good sleep reduces risk of return to use | I think it's harder to stay clean and sober when you're not at your best and you're not rested and you're not feeling good. Because as soon as you start feeling bad, you get triggered and those are the times when you're the most vulnerable to use again, and so if you're not getting sleep, then you're not gonna be able to handle those triggers as well, like if you were to run into somebody on the street that you used to use with or something like that. If you're not sleeping well and you're tired you might not be as like strong I guess to, you know, turn down the offer to get high." -Study ID 021 |
| Buprenor-phine's role in sleep | Buprenorphine impacts energy levels for sleep | "I would say sometimes I notice if I don't get a lot of sleep after I take my morning dose, I'm like, I want to go straight back to sleep. My eyes get very heavy or sleepy. But that doesn't really, it's not like a high, it's just I need to go close my eyes." -Study ID 046 |
| | Buprenorphine affects mental and physical health and subsequently sleep | 'I'm thankful for buprenorphine because it has allowed me time to address some sleep behaviors and habits and stuff, and change those things, work on changing, instead of just going to use." -Study ID 045 |
| | Buprenorphine schedules relate to sleep | "If you don't sleep at night, then when you take it in the morning, and it's kind of you know, makes you groggy. But if you got a good sleep, then you know, it kind of helps you prepare for your day." -Study ID 14 |

about the challenges of sleep while in early recovery, with insufficient sleep and 'tossing and turning' and dreams about active addiction. Women described the transition of early recovery into long-term recovery, with several women describing how getting into a routine with treatment (n=5) helped stabilize sleep schedules.

Women also described how good sleep reduces risk of return to use. Participants reported this relationship through several different mechanisms. Participants described a direct link between sleep and return to use, noting that if they get proper sleep, they do not experience cravings to use. Some participants indicated more underlying mechanisms; for example, one participant shared that being awake at night can lead to thinking negative thoughts about returning to use, and that such negative self-talk can contribute to their return to use. Other

participants (n=2) noted that insufficient sleep can contribute to lack of impulse control, which could facilitate a return to use.

**Theme 3: Buprenorphine's role in sleep.** Although participants shared about the stabilization of their sleep while in long-term recovery, they shared mixed experiences with buprenorphine's role in impacting their sleep. Some participants reported that buprenorphine has helped with their sleep, noting that it also helps manage pain. Other participants shared that they did not feel that buprenorphine has impacted their sleep. A few participants reflected that buprenorphine would cause drowsiness, which helped with sleep (n=3).

However, other participants (n=2) reported seemingly opposite effects, noting that buprenorphine would create a lot of energy, particularly at the beginning of recovery. For these reasons, some participants found it helpful to time their buprenorphine dosing with when they wanted to be alert, while others preferred to time their dosing when they wanted to be drowsy for sleep. Participants noticed that sleep and buprenorphine schedules could also be a bi-directional relationship, such that lack of sleep might lead to forgetfulness, and thus forgetting to take their buprenorphine.

## 4. Discussion

To our knowledge, this study is among the first to explore patient-reported experiences with sleep and recovery among women receiving buprenorphine with insomnia symptoms. We combined survey data with rich qualitative data from interviews to achieve a comprehensive understanding of women's perspectives of sleep during OUD treatment. We found that sleep in recovery is a bidirectional and multifaceted process among this sample. Women receiving MOUD described sleep as being important to them and their recovery. They described qualities of healthy sleep and factors that impact sleep, including buprenorphine. These data are patient-derived and contributes to a limited evidence base of subjective experiences with sleep among people with OUD.

In one qualitative study, people with a history of heroin use indicated that sleep problems posed a barrier to recovery and improved quality of life [39]. These findings were reflected in another study that demonstrated that among people in treatment for polysubstance use, sleep issues were associated with distress; establishing healthy sleep behaviors, including implementing routines, helped participants address such issues in recovery [40]. Similarly, women quantitatively described sleep health behaviors, both positive and negative, that they engaged in that impacted their sleep. Qualitatively, women elucidated the context behind such behaviors by describing the cyclical nature of the multidimensional factors that impact sleep and how these factors serve as the underlying mechanisms between how sleep then impacts OUD.

Although this bi-directional relationship between health behaviors and sleep has been demonstrated to worsen health outcomes in the general population, it has particular impact for populations with OUD as it may affect treatment outcomes [41]. In our sample, women often reported mental health symptomology interfering with sleep; this aligns with recent research demonstrating that people on MOUD with insomnia symptoms may experience greater levels of depression, anxiety, and post-traumatic stress than those without [8,42]. This is particularly salient given that literature demonstrates that co-morbid mental health conditions and sleep issues may impact the effectiveness of substance use treatment; for example, one study among treatment-seeking individuals with SUD found that mental health conditions actually mediated the relationship between severity of substance use and insomnia [43]. Addressing both mental health issues and insufficient sleep in treatment is a promising avenue to addressing treatment outcomes; this intersection provides an exciting opportunity for interventions such as a recently developed multi-disciplinary therapy to both reduce anxiety

and improve sleep health among adults in treatment for substance use disorders [41]. Thus, targeting other highly comorbid conditions such as anxiety and depression in this population might additionally address sleep outcomes [44].

Many of the women reported how social determinants impacted their sleep. For example, they reported how caregiving and physical environment impacted their ability to fall and stay asleep, and how well they could fully engage in healthy sleep behaviors. This is reflected in existing literature; Neale and colleagues found that among people in recovery from SUDs, participants also reported social and environmental barriers to sleep [45]. This is salient as people with OUD are likely to experience negative social determinants of health, and prior research has demonstrated that such determinants may preclude women with OUD from participating in behavioral sleep interventions [35,46]. These findings indicate that upstream factors should be considered in the development, study, and evaluation of interventions (behavioral and pharmacologic) aimed to address sleep in women with OUD.

In addition, such research needs to incorporate the patient voice, such as with a qualitative understanding about experiences with sleep that are not captured in traditional quantitative measures, to optimize equity in outcomes as new therapeutics transition across the translational science pipeline [47]. Strong qualitative data has the potential to directly inform sleep interventions in the clinical setting; for example, by helping inform tailored sleep assessments into regular MOUD visits or training behavioral clinicians on sleep among women on MOUD. Further, the relationship of upstream factors to sleep in this population poses an opportunity for the development of multidisciplinary interventions integrating different clinical fields.

Many women receiving MOUD insightfully identified lack of sleep as a potential contributing factor to return to use. This finding aligns with existing research, with one study noting that among their sample, people on MOUD with insomnia were more likely than those without insomnia to report that poor sleep was interfering with their MOUD treatment, and that improved sleep would help them achieve their treatment goals [8]. Together, these findings reflects literature that has found that insufficient sleep can compromise OUD treatment outcomes via multiple underlying mechanisms, such as increased impulsivity, heightened stress, distress tolerance, and pain [13,48]. Even though sleep disturbances among people with OUD have historically been largely attributed to ongoing substance use, recent literature highlights how insomnia symptoms persist throughout all stages of OUD recovery, but may take different shape as recovery evolves [7,13,35,49]. Thus, these findings reflect how insufficient sleep is likely not simply a manifestation of OUD itself but an inherent component of a phenotypic makeup of individuals with OUD, further supporting how sleep health is a promising target for personalized interventions, including for individuals engaged in MOUD treatment [11].

Existing qualitative data on sleep in people with substance use disorders has been primarily collected in broad substance use populations or in inpatient settings. To our knowledge, no published research exists on patient-reported sleep experiences for women in long-term recovery and outpatient maintenance on buprenorphine. This qualitative data provides rich context that is otherwise not captured by brief self-reported sleep measures. For example, women reported conflicting perspectives on the effects of buprenorphine on their sleep; some women reported that buprenorphine helped energize them, others reported that it made them sleepy, and yet others shared that it did not impact their sleep. These varying experiences are unsurprising, since the physiological effects of different substances may vary person-to-person [45,50].

However, the nuance to which people experience the effects of buprenorphine that is captured in this qualitative data highlights the limitations of interventions that do not incorporate a personalized medicine approach, where treatment regimens are individualized based on

patients' needs. Our findings provide a lived experience context for future sleep-OUD studies to incorporate into their designs, such as by using the patient perspective to inform outcome definitions (e.g., goals for healthy sleep) and selection of multi-level covariates that can impact sleep (e.g., timing of buprenorphine). In addition, research questions generated by this manuscript may be followed up by future studies that utilize more standardized sleep scales.

## 4.1  Limitations

There are limitations to our study. Our sample had been on buprenorphine for variable amounts of time, ranging from 2 months to 20 years. Thus, we were not able to robustly explore sleep experiences in people in short vs. long term recovery, which provides an avenue for research into perspectives on differences between sleep at varying stages of recovery. In addition, we did not conduct member checking or other procedures to reflect the findings back to the qualitative participants to ensure that our findings accurately represented their experiences. Finally, this study included women on MOUD from one clinic, which limits generalizability of findings to male patients, patients from other clinics, or patients with other substance use disorders. However, our findings help elucidate sleep health in this particular population, and their perspectives allow us to identify underlying mechanisms between sleep and recovery to develop holistic and multi-disciplinary sleep treatments to accompany MOUD.

## 5.  Conclusion

We found that sleep is a multidimensional and dynamic process among this sample of women on MOUD. To our knowledge, this study is among the first to evaluate sleep experiences among women on MOUD, and can help inform tailored, gender-responsive sleep interventions among this population. Incorporating these experiences into the study, development, and implementation of inclusive sleep treatments for OUD patients will help ensure that, as new science at the OUD-sleep intersection emerges, its ultimate translation into clinical care is capable of improving patient outcomes.

## Supporting information

**S1 Text.  Interview Guide. Contains qualitative interview guide.**
(DOCX)

## Disclosure Statement

There are no relevant financial or non-financial competing interests to report.

## Author contributions

**Conceptualization:** Joseph M. Dzierzewski, Caitlin E. Martin.

**Data curation:** Michelle Eglovitch, Stephanie Violante.

**Formal analysis:** Michelle Eglovitch, Alison J. Patev, Brenna Cook, Chengxian Shi.

**Funding acquisition:** Caitlin E. Martin.

**Investigation:** Joseph M. Dzierzewski, Caitlin E. Martin.

**Methodology:** Joseph M. Dzierzewski, Caitlin E. Martin.

**Project administration:** Michelle Eglovitch, Stephanie Violante, Caitlin E. Martin.

**Resources:** Caitlin E. Martin.

**Supervision:** Caitlin E. Martin.

**Writing – original draft:** Michelle Eglovitch.

**Writing – review & editing:** Anna Beth Parlier-Ahmad, Alison J. Patev, Brenna Cook, Chengxian Shi, Stephanie Violante, Joseph M. Dzierzewski, Morgan H. James, Caitlin E. Martin.

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
