## [Decision Letter · Decision Letter 0]

14 Nov 2024

PMEN-D-24-00493

Perspectives and experiences with sleep and recovery among women receiving buprenorphine for opioid use disorder

PLOS Mental Health

Dear Dr. Eglovitch,

Thank you for submitting your manuscript to PLOS Mental Health. After careful consideration, we feel that it has merit but does not fully meet PLOS Mental Health’s publication criteria as it currently stands. Therefore, we invite you to submit a revised version of the manuscript that addresses the points raised during the review process.

We look forward to receiving your revised manuscript.

Kind regards,

Rujing Zha

Academic Editor

PLOS Mental Health

Journal Requirements:

Additional Editor Comments (if provided):

Reviewers' comments:

Reviewer's Responses to Questions

**Comments to the Author**

1. Does this manuscript meet PLOS Mental Health’s publication criteria ? Is the manuscript technically sound, and do the data support the conclusions? The manuscript must describe methodologically and ethically rigorous research with conclusions that are appropriately drawn based on the data presented.

Reviewer #1: Yes

Reviewer #2: Partly

Reviewer #3: Partly

Reviewer #4: Partly

Reviewer #5: Yes

Reviewer #6: Yes

2. Has the statistical analysis been performed appropriately and rigorously?

Reviewer #1: No

Reviewer #2: No

Reviewer #3: No

Reviewer #4: No

Reviewer #5: No

Reviewer #6: Yes

3. Have the authors made all data underlying the findings in their manuscript fully available (please refer to the Data Availability Statement at the start of the manuscript PDF file)?

Reviewer #1: No

Reviewer #2: No

Reviewer #3: No

Reviewer #4: Yes

Reviewer #5: No

Reviewer #6: Yes

4. Is the manuscript presented in an intelligible fashion and written in standard English?

Reviewer #1: Yes

Reviewer #2: Yes

Reviewer #3: Yes

Reviewer #4: Yes

Reviewer #5: Yes

Reviewer #6: Yes

5. Review Comments to the Author

Reviewer #1: This manuscript describes a crucial research topic: insomnia in drug recovery. Indeed, once this topic has been reviewed, it will provide information needed by the academic literature on considering perspectives of users of drugs since they are the one experience it and felt the need to recover. they are expert of their problems and may provide professionals other ways of helping them to recover.Focusing on the methodology where the authors say that the design is mixed, but it is noticeable that it is more qualitative. I encourage them to develop the quantitative aspect as well so that it fulfills the elements of a mixed approach.

Reviewer #2: This is a valuable research study that contributes to the advancement of sleep medicine knowledge. However, to further enhance its impact, consider addressing the following points:

Methodology:

Scale Validation: Please clarify whether the Scale for Sleep Behaviors was validated for the specific population and context of this study.

Interview Guide: Providing the interview guide would enhance transparency and reproducibility.

Interview Methods: While Zoom interviews were mentioned, please specify if other modes (in-person or phone calls) were used. If so, please detail the recording and transcription processes for these different methods.

Data Analysis: Please provide a more detailed explanation of the statistical analyses performed on the quantitative data.

Data Sharing: While the transcripts were anonymized, sharing the coded transcripts (without personal identifiers) could further contribute to the transparency and replicability of the study.

By addressing these points, you can strengthen the rigor and impact of your research.

Reviewer #3: Thank you for the opportunity to review this manuscript. The topic is both exciting and unique, addressing an understudied area of sleep health in women undergoing OUD treatment with MOUD. The manuscript effectively combines both quantitative and qualitative methodologies, providing a rich and multifaceted understanding of how sleep impacts recovery outcomes. However, there is much room for improvement in the manuscript. Please check my specific and overall comments.

Abstract: The abstract provides a good foundation but would benefit from a few enhancements to strengthen its clarity and impact. I suggest incorporating more specific findings in the results section. Additionally, clarifying the effects of OUD medication on sleep quality, particularly any challenges or benefits reported by participants, would enrich the interpretation. Also, a policy-level recommendation would offer a meaningful contribution to policy discussions.

Introduction

Lines 97–99: Although comorbid psychological conditions are highlighted as factors in dropout rates, further clarification on the prevalence and specific impact of sleep disorders as comorbidities in previous research would strengthen the connection to this study's focus.

Lines 101–110: The link between poor sleep and substance use recurrence could be expanded to include more recent evidence or direct statistics on how sleep issues impact relapse, specifically within the OUD population. This would highlight the need for sleep interventions in OUD treatment.

Lines 121–130: The justification for using a qualitative approach is sound but could be more explicit about why this approach is particularly critical for understanding patient-centered experiences in this context. Highlighting that qualitative insights can reveal treatment gaps or inform targeted sleep interventions for women would provide more substantial support.

Lines 131–138: While the gap in research on sleep and OUD recovery from the patient perspective is noted, the section could more clearly state the specific need for patient-reported outcomes on sleep during OUD recovery, linking this to how it can inform therapeutic adjustments.

Methods

-Overall comment: The methods section is well-detailed, offering a comprehensive description of participant recruitment, data collection, and both qualitative and quantitative measures used to assess sleep, mental health, and other psychosocial factors among women in MOUD treatment. However, the study could be enriched by including statistical tests, such as a chi-squared test, to explore significant associations between clinically elevated insomnia symptoms and other demographic or psychosocial variables. This addition would provide a clearer quantitative insight into factors associated with insomnia severity

Result

Line 277-228: There's an inconsistency in reporting the number of participants with clinically elevated insomnia symptoms. The methods section states that 54 participants reported elevated insomnia symptoms, but the results section describes only 50 participants meeting this criterion. This discrepancy suggests a possible oversight or error in participant tracking or data reporting. The researchers may have unintentionally excluded four participants or may not have updated the sample size accurately between sections. Clarifying this inconsistency would be essential to ensure the integrity and reliability of the study findings. The researchers should explain any exclusion criteria or adjustments made after the initial count to avoid confusion and provide transparency in participant selection.

Line 283-284: According to the STROBE guidelines, "Sociodemographic Characteristics" should indeed refer to the full distribution of all participants, which in this study is n=78n = 78n=78. If the researchers intend only to present characteristics of participants with clinically elevated insomnia symptoms, the table title should clarify this scope to avoid confusion. A clearer title could be: "Table 1: Sociodemographic Characteristics Among Respondents with Clinically Elevated Insomnia Symptoms." This approach maintains transparency and aligns with research reporting standards, ensuring readers understand which subset of the study population is being described.

Table 1: Also, in Table 1, there are a few discrepancies that raise questions about the data's accuracy and the methodology used. First, there is an issue with the Race category. The percentages for race among participants with clinically elevated insomnia symptoms sum to 100.9% (64.0% White, 30.0% Black, and 6.9% Other), which exceeds 100% and is not feasible, suggesting a potential calculation or rounding error. Additionally, the Employment status category also exhibits an irregularity: the number of participants classified as unemployed (50.0%) and employed (30.0%) only accounts for 80% of the total sample with elevated insomnia symptoms, leaving 20% unaccounted for, which could imply missing data or another calculation error.

Line 288: "Co-morbid anxiety and depression were highly prevalent in the overall sample, with nearly half of the participants endorsing clinically significant anxiety, and almost 3/4ths of the sample endorsing clinically significant depression." Here, the authors gave a vague interpretation. The authors provide a general statement on the prevalence of anxiety and depression but do not clarify the true percentage of participants experiencing both conditions concurrently. Calculating and reporting the percentage of respondents with co-morbid anxiety and depression would offer a more precise measure of co-morbidity, allowing for clearer insight into the overlap of these symptoms within the sample.

Line 290-291: "The average buprenorphine dose was 24mg/day, which is within the standard range of buprenorphine dosing." The authors presented the median value in Table 2, “Buprenorphine dose (milligrams) (median [range]) 24 (4-30)”, but they interpreted it as a mean value, which is totally incorrect.

Discussion

-Overall comment: The discussion points out that the study is among the first to examine sleep experiences, specifically among women on buprenorphine. This could be emphasised more prominently at the start of the discussion to underscore the study’s unique contributions to both the sleep and OUD treatment fields. Framing the findings as filling a crucial gap in gender-specific research on sleep during OUD recovery would also strengthen its significance.

-The discussion section could be significantly strengthened by incorporating more evidence to back up the study's findings, particularly regarding the implications of sleep on recovery outcomes and the specific role of MOUD in influencing sleep patterns among individuals with OUD.

Line 96-99: The discussion mentions that sleep, anxiety, and depression are interlinked and can influence treatment outcomes in OUD. However, evidence detailing how anxiety and depression specifically mediate the impact of poor sleep on recovery would clarify this relationship. Including studies that analyse the impact of co-morbid mental health conditions on sleep quality and recovery outcomes among MO.

Line 419-428: While the discussion suggests that sleep health is a promising target for interventions, it could be further developed by proposing specific clinical or policy recommendations. For instance, integrating sleep health assessments into routine OUD treatment or training clinicians on sleep hygiene for patients could be practical applications of this research. Expanding on the potential for multidisciplinary interventions to address sleep disturbances, particularly for women balancing caregiving and environmental stressors, would enhance the discussion’s relevance for practice.

Line 460-471: The manuscript would benefit from a clearer presentation of its strengths and limitations. Consider adding a dedicated bullet-point to distinguish this section.

Line 473-477: The manuscript could conclude with a more definitive statement about the importance of sleep in OUD recovery, emphasising the potential benefits of addressing sleep in OUD interventions. A strong closing statement highlighting the promise of targeted, gender-responsive sleep interventions would leave a lasting impression and reinforce the study’s contribution to improving outcomes for women in MOUD.

Reviewer #4: It is mentioned that the study corresponds to a mixed method, it is inferred that it is sequential, but it is not explicitly stated what type of study it corresponds to.

They mention the instruments used in the quantitative phase and state that they are valid, but do not provide specific data on their psychometric properties to justify their use in comparison with other existing instruments.

Some bivariate analyses can be carried out to show whether the secondary variables are related to the main one.

In the data analysis (interpretation) phase, there is no clear description of the data integration phase.

Reviewer #5: General Comments

Congratulations to authors, for this great work

Abstract

Can you add a line on when was the study done, and how you recruited the participants.

Methods

Be more specific how recruitment for semi-structured interviews was done and the settings of the interview. When and where was the interviews conducted, for how long, how many interviews was done to get the saturation point?

Which data were obtained from the medical reports? (line 173-174)

Are you measurement tools validated for your study population?

What was the objectives of your study? with respect to use of GAD-7 and PHQ-9? Why did you opt to use GAD-7 and PHQ-9 to screen for sleep pattern?

GAD-7 and PHQ-9 screen for anxiety and depression symptoms which if I understand is not the aim of your study as per your title and introduction. Do you need to review your title and introduction to rationalize the use of these two tools?

Results

Your first paragraph of results, line 277-278 contradicts with Line 175-177 of the methodology part. Be specific how many were included in analysis 54 or 50?

NB: You have used a lot of tools like GAD-7 and PHQ-9 but not analyzed, what is the rationale of using a tool and not analyzed?

Reviewer #6: The study addresses an important and underexplored topic: the relationship between sleep issues and opioid use disorder (OUD) recovery among women. The use of a mixed-methods design is appropriate and commendable, as it captures both quantitative and qualitative data, providing a richer understanding of participants' subjective sleep experiences. The study findings have the potential to inform clinical practices aimed at supporting women with OUD by addressing sleep-related challenges.

6. PLOS authors have the option to publish the peer review history of their article (what does this mean? ). If published, this will include your full peer review and any attached files.

**Do you want your identity to be public for this peer review?** For information about this choice, including consent withdrawal, please see our Privacy Policy .

Reviewer #1: **Yes: ** Emmanuel Mihigo M.

Reviewer #2: No

Reviewer #3: No

Reviewer #4: **Yes: ** Sandra Paola Mondragón Bohórquez

Reviewer #5: No

Reviewer #6: **Yes: ** Alberto Muanido

---

## [Editor Report · Decision Letter 1]

23 Dec 2024

PMEN-D-24-00493R1

Perspectives and experiences with sleep and recovery among women receiving buprenorphine for opioid use disorder

PLOS Mental Health

Dear Dr. Eglovitch,

Thank you for submitting your manuscript to PLOS Mental Health. After careful consideration, we feel that it has merit but does not fully meet PLOS Mental Health’s publication criteria as it currently stands. Therefore, we invite you to submit a revised version of the manuscript that addresses the points raised during the review process.

We look forward to receiving your revised manuscript.

Kind regards,

Rujing Zha

Academic Editor

PLOS Mental Health

Journal Requirements:

1. Please describe in your methods section how capacity to provide consent was determined for the participants in this study. Please also state whether your ethics committee or IRB approved this consent procedure. If you did not assess capacity to consent please briefly outline why this was not necessary in this case.
---

## [Decision Letter · Decision Letter 2]

4 Feb 2025

PMEN-D-24-00493R2

Perspectives and experiences with sleep and recovery among women receiving buprenorphine for opioid use disorder

PLOS Mental Health

Dear Dr. Eglovitch,

Thank you for submitting your manuscript to PLOS Mental Health. After careful consideration, we feel that it has merit but does not fully meet PLOS Mental Health’s publication criteria as it currently stands. Therefore, we invite you to submit a revised version of the manuscript that addresses the points raised during the review process.

We look forward to receiving your revised manuscript.

Kind regards,

Rujing Zha

Academic Editor

PLOS Mental Health

Journal Requirements:

Additional Editor Comments (if provided):

Reviewers' comments:

Reviewer's Responses to Questions

**Comments to the Author**

1. If the authors have adequately addressed your comments raised in a previous round of review and you feel that this manuscript is now acceptable for publication, you may indicate that here to bypass the “Comments to the Author” section, enter your conflict of interest statement in the “Confidential to Editor” section, and submit your "Accept" recommendation.

Reviewer #2: (No Response)

Reviewer #3: All comments have been addressed

Reviewer #4: All comments have been addressed

Reviewer #5: All comments have been addressed

2. Does this manuscript meet PLOS Mental Health’s publication criteria ? Is the manuscript technically sound, and do the data support the conclusions? The manuscript must describe methodologically and ethically rigorous research with conclusions that are appropriately drawn based on the data presented.

Reviewer #2: Yes

Reviewer #3: Yes

Reviewer #4: Yes

Reviewer #5: Yes

3. Has the statistical analysis been performed appropriately and rigorously?

Reviewer #2: Yes

Reviewer #3: Yes

Reviewer #4: Yes

Reviewer #5: Yes

4. Have the authors made all data underlying the findings in their manuscript fully available (please refer to the Data Availability Statement at the start of the manuscript PDF file)?

Reviewer #2: Yes

Reviewer #3: No

Reviewer #4: Yes

Reviewer #5: Yes

5. Is the manuscript presented in an intelligible fashion and written in standard English?

Reviewer #2: Yes

Reviewer #3: Yes

Reviewer #4: Yes

Reviewer #5: Yes

6. Review Comments to the Author

Reviewer #2: Nice revision to the previous manuscript.

My suggestions are really minor

When enrolling the participants, one of the exclusion criteria was that they were unable to complete the informed consent procedure. Does this mean these had passed the cognitive and psychiatric evaluation? If they had and eventually accepted to take part in the consent process but later declined, what was done to them afterwards

This may be minor. In the methods section line 182, you indicate 'see below'. It would be helpful if you guided the reader where to check for the said reference either specify the subheading.

Under the results section, there is a caption: 'Figure 1: Healthy Sleep Behaviors'. However there is no such a graph or any image accompanying the caption. Consider adding this.

Reviewer #3: Authors met my recommendations. Good luck.

Reviewer #4: The authors have generally responded satisfactorily to the suggestions. However, it is important that the methodology explicitly states how the integration of quantitative and qualitative data has been carried out.

Reviewer #5: The manuscript is now suitable for publication

7. PLOS authors have the option to publish the peer review history of their article (what does this mean? ). If published, this will include your full peer review and any attached files.

**Do you want your identity to be public for this peer review?** For information about this choice, including consent withdrawal, please see our Privacy Policy .

Reviewer #2: No

Reviewer #3: No

Reviewer #4: No

Reviewer #5: No

---

## [Decision Letter · Decision Letter 3]

3 Mar 2025

Perspectives and experiences with sleep and recovery among women receiving buprenorphine for opioid use disorder

PMEN-D-24-00493R3

Dear Ms. Eglovitch,

We are pleased to inform you that your manuscript 'Perspectives and experiences with sleep and recovery among women receiving buprenorphine for opioid use disorder' has been provisionally accepted for publication in PLOS Mental Health.

Best regards,

Rujing Zha

Academic Editor

PLOS Mental Health

Reviewer Comments (if any, and for reference):

Reviewer's Responses to Questions

**Comments to the Author**

1. If the authors have adequately addressed your comments raised in a previous round of review and you feel that this manuscript is now acceptable for publication, you may indicate that here to bypass the “Comments to the Author” section, enter your conflict of interest statement in the “Confidential to Editor” section, and submit your "Accept" recommendation.

Reviewer #2: All comments have been addressed

Reviewer #4: All comments have been addressed

2. Does this manuscript meet PLOS Mental Health’s publication criteria ? Is the manuscript technically sound, and do the data support the conclusions? The manuscript must describe methodologically and ethically rigorous research with conclusions that are appropriately drawn based on the data presented.

Reviewer #2: Yes

Reviewer #4: Yes

3. Has the statistical analysis been performed appropriately and rigorously?

Reviewer #2: Yes

Reviewer #4: Yes

4. Have the authors made all data underlying the findings in their manuscript fully available (please refer to the Data Availability Statement at the start of the manuscript PDF file)?

Reviewer #2: Yes

Reviewer #4: Yes

5. Is the manuscript presented in an intelligible fashion and written in standard English?

Reviewer #2: Yes

Reviewer #4: Yes

6. Review Comments to the Author

Reviewer #2: Well done

Reviewer #4: It´s a interesting paper, all corrections have been made

7. PLOS authors have the option to publish the peer review history of their article (what does this mean? ). If published, this will include your full peer review and any attached files.

**Do you want your identity to be public for this peer review?** For information about this choice, including consent withdrawal, please see our Privacy Policy .

Reviewer #2: No

Reviewer #4: No
